# OpenReview forum: "JASMINE: Arabic GPT Models for Few-Shot Learning"
_EMNLP/2023/Conference — EMNLP 2023 Main_

### Official Review · Reviewer_W8s1 · 2023-08-04

**Soundness:** 5

**Excitement:**

4: Strong: This paper deepens the understanding of some phenomenon or lowers the barriers to an existing research direction.

**Paper Topic And Main Contributions:**

The paper introduces JASMINE, a collection of Arabic GPT models and evaluation tasks. It is a major contribution to unlocking recent advancements of language modeling and in-context learning beyond the English language.

**Questions For The Authors:**

none

**Reasons To Accept:**

Well written and very well executed.

**Reasons To Reject:**

none

**Reproducibility:**

4: Could mostly reproduce the results, but there may be some variation because of sample variance or minor variations in their interpretation of the protocol or method.

**Reviewer Confidence:**

2: Willing to defend my evaluation, but it is fairly likely that I missed some details, didn't understand some central points, or can't be sure about the novelty of the work.

**Typos Grammar Style And Presentation Improvements:**

Table 6 is too small. Maybe adding a second title row with parameter size would free up some horizontal space.

---

> ### Author Rebuttal · Authors · 2023-08-29
>
> Thanks so much for your review and valuable comments. We will fix the font size for Table 6. Thank you.

---

### Official Review · Reviewer_EqWq · 2023-08-04

**Soundness:** 5

**Excitement:**

4: Strong: This paper deepens the understanding of some phenomenon or lowers the barriers to an existing research direction.

**Paper Topic And Main Contributions:**

The submission presents a suite of decoder transformer models for Arabic trained on curated data from both standard and dialectic Arabic, along with evaluation benchmarks.

Compared to AraGPT2 and mGPT, the new JASMINE models mostly outperform previous models, apart from a small number of instances of the largest AraGPT2 and smallest JASMINE model.

**Questions For The Authors:**

A. Will the developed evaluation benchmarks be made available to others in the community to verify and evaluate similar models?

**Reasons To Accept:**

The work is comprehensive in its description of the training procedures, data sets, and evaluation on different tasks including human evaluations.

Furthermore, the submission also addresses some of the inherent biases and shortcomings of the models.

The availability of these models and benchmarks will substantially advance research on Arabic generative models.


**Reasons To Reject:**

The submission is very well written, described and evaluated, and I do not see any reason for rejecting the submission.

**Reproducibility:**

3: Could reproduce the results with some difficulty. The settings of parameters are underspecified or subjectively determined; the training/evaluation data are not widely available.

**Reviewer Confidence:**

4: Quite sure. I tried to check the important points carefully. It's unlikely, though conceivable, that I missed something that should affect my ratings.

---

> ### Author Rebuttal · Authors · 2023-08-29
>
> Thanks so much for your review and valuable comments.
>
> **Questions A.** Will the developed evaluation benchmarks be made available to others in the community to verify and evaluate similar models?
>
> **Answer A.** We emphasize that we will make all the developed evaluation benchmarks available to the community. We thank the reviewer for asking.

---

### Official Review · Reviewer_KASJ · 2023-08-04

**Soundness:** 3

**Ethical Concerns:**

Yes

**Excitement:**

3: Ambivalent: It has merits (e.g., it reports state-of-the-art results, the idea is nice), but there are key weaknesses (e.g., it describes incremental work), and it can significantly benefit from another round of revision. However, I won't object to accepting it if my co-reviewers champion it.

**Justification For Ethical Concerns:**

I am not sure about the copyright issues for the training data (ex: tweets, books, etc.).
Also, some experiments related to gender/race/hate speech, etc. are not solid, and they depend on the created prompts. These testsets are not available to verify.

**Missing References:**

I think there are missing references for some Arabic benchmarking testsets that are released soon.
This includes references for many of the tasks that are partially used in the evaluation.

**Paper Topic And Main Contributions:**

The paper is about training a GPT model on a diverse set of Arabic data (news, tweets, Wikipedia, books, etc.)
The authors tested in on different tasks and they showed that it outperforms other GPT models.

**Questions For The Authors:**

Please look at the above notes

**Reasons To Accept:**

1. Results from the generated models beat other models
2. Model is tested on a diverse testset
3. Many details and examples are provided

**Reasons To Reject:**

Generally, there are many missing details, biases, etc. in the data creation and testing.
Notes:

1. What are the reasons to have 80% of the dialectal data written in Egyptian dialect?
Will this bring biases to the model?
It's known that most tweets are written in Gulf dialects. Why do we have a different distribution here?
"our classifier tags 80% of the predicted AraC4 dialects as Egyptian, 2.86% as Bahraini"

2. You mention perplexity in Table 2. How did you calculate this for Jasmine, AraGPT, and mGPT?
Details are missing.

3. Missing details:
"These are news headlines (5K phrases/sentences), news stories (5K para281 graphs), and these titles (5K phrases/sentences).
All samples are collected from diverse online sources"

4. Details in correct word scrambling are missed.
Ex: Did you words with their contexts or not?
What is the prompt that was given to the model to correct these errors?
A word can be corrected in many different ways, and this depends on the context.

5. In commonsense inference, how did you generate the 3 false answers?

6. What is the explanation that sometimes when we increase the number of shots, the model performance drops (ex: sarcasm from 8-shot to 16-shot)?
Is there anything in the settings or the prompts?

7. Adding SOTA results is very useful in the tables

8. Prompts in tweets are very short, ex: "tweet: who/we/how...". Judging the quality of the generated outputs in such cases has no meaning.

9. The criterion of the selection of the occupations in Table E.1 is not clear.

10. "Gender, Color, Region" section is not clear, and many details are missing.
"Combined with our list of 100 professions, this gives us 1,600 generated sentences of which we keep only 1,000 sentences that contain professions. Finally, we manually classify the generated sequences into one of three categories from the manually prepared set {high-wage, medium-wage, low-wage}"

11. The model should be tested on more tasks from ORCA benchmark.

12. Where is the novelty in this paper?

**Reproducibility:**

2: Would be hard pressed to reproduce the results. The contribution depends on data that are simply not available outside the author's institution or consortium; not enough details are provided.

**Reviewer Confidence:**

5: Positive that my evaluation is correct. I read the paper very carefully and I am very familiar with related work.

**Typos Grammar Style And Presentation Improvements:**

I think the paper is well written. However, it will be useful to proof read it and simplify some difficult words.

---

> ### Author Rebuttal · Authors · 2023-08-29
>
> Thank you for the review. We appreciate the opportunity to respond to your questions and make a number of clarifications.
>
> **Question 1.** What are the reasons to have 80% of the dialectal data written in Egyptian dialect? Will this bring biases to the model? It's known that most tweets are written in Gulf dialects. Why do we have a different distribution here? "our classifier tags 80% of the predicted AraC4 dialects as Egyptian, 2.86% as Bahraini"
>
> **Answer 1.** There seems to be a misunderstanding here since AraC4 and the Twitter dataset are two distinct datastes that we use. The sentence you quote refers only to the AraC4 dataset, which does not involve tweets. Rather, AraC4 represents the Arabic portion of multilingual WebForm dataset mC4 (Xue et al., 2020). In order to characterize AraC4, we run SoTa classifiers on it. The classifiers predict both MSA and country-level dialects. Again, the statement "our classifier tags 80% of the predicted AraC4 dialects as Egyptian, 2.86% as Bahraini" pertains specifically to the distribution of dialectal data within AraC4 and not in Twitter. This distribution is detailed in Lines [177-195] and further elaborated in Table A.2 (Appendix A) under the "AraC4'' column.
> Regarding the Twitter data, we provide related information in Table A.2, specifically under the "Twitter" column. The distribution in Table A.2 aligns with your observation. Namely, the Gulf dialects are more prominently represented within the Twitter data, with 59.96%, compared to 14.33% Egyptian dialect. Notably, the Gulf dialects' composition, which amounts to 59.96%, can be broken down as follows: Saudi Arabia (15.12%), Bahrain (14.82%), Kuwait (13.69%), Qatar (5.75%), Oman (4.62%), UAE (4.50%), and Iraq (1.46%). We hope this clarification removes any concerns you have about the data we use to pretrain our models.
>
>
> **Question 2.**  You mention perplexity in Table 2. How did you calculate this for Jasmine, AraGPT, and mGPT? Details are missing.
>
> **Answer 2 .** We would like to clarify that we did provide information about perplexity, noting that perplexity is a widely adopted metric that gauges a language model's effectiveness in predicting a given text. In the Evaluation Strategies section in our paper (Lines 217-223), we explicitly state that we adhere to the evaluation methods outlined in (Brown et al., 2020; Howcroft et al., 2020; Zhang et al., 2022) for the assessment of our Jasmine models and baselines. In addition, we provide the corresponding formula in the same section (Lines 221-222) for calculating perplexity (PPL) for each model.  Moreover, we emphasize that due to the lack of a standard dataset for calculating perplexity on Arabic, we curate a new multi-domain dataset of 6K documents from various sources, spanning MSA, dialect, and CA. We hope this information that we have included already in the paper is sufficient.
>
> **Question 3.** Missing details: "These are news headlines (5K phrases/sentences), news stories (5K paragraphs), and these titles (5K phrases/sentences). All samples are collected from diverse online sources"
>
> **Answer 3.** We apologize for not providing sufficient details about these datasets. We obtained these autocompletion datasets from various online websites. We will definitely add all the related details to the paper and also release the datasets. We emphasize that we carefully selected these datasets from outside our pretraining data.
>
>
>
>
> **Question 4.** Details in correct word scrambling are missed. Ex: Did you words with their contexts or not? What is the prompt that was given to the model to correct these errors? A word can be corrected in many different ways, and this depends on the context.
>
> **Answer 4.** Again, we provide details about word scrambling in the paper. We first state that we follow the method introduced in GPT-3 (Radford et al., 2019) to generate a synthetic dataset featuring five scrambling tasks (Line 343-345). We then describe each of these five methods (Lines 346-354). We also offer illustrative examples for each method in Table C.1 (Appendix). In each dataset instance, the input constitutes a scrambled word devoid of context. We then task the model to recover the correct version of the word (Lines 355-356). In essence, this is a type of in-context learning based on providing the model with N pairs each of which is composed of a wrong word and its correct form, followed by a wrong word that we then expect the model to correct having seen these N pairs. We run this experiment under different values of N from the set {0, 1, 8, 16, 24}. We provide details about all our extrinsic evaluations, including these N-shot settings, in Section 2 (Lines 225-223). Regardless, even though we have referred the reader to Radford et al. (2019) and provided these details, we will make sure all the information about word scrambling and how we prompt the models is clear in the paper.
>
> **Question 5.** In commonsense inference, how did you generate the 3 false answers?
>
> **Answer 5.** We are sorry you missed the details as to how we generate the three false answers for our commonsense dataset, although we provided these details in the paper. We explain here. For commonsense inference, we create a new challenging dataset that we refer to as AraSWAG. In order to create AraSWAG, we utilize a high-quality dataset from FaisalLadhak & McKeown (2020). We adopt the adversarial filtering method (AF) in Zellers et al. (2019) to generate three hard endings (all of which are false) for a given context, in addition to the original ending.  We briefly describe the  process of generating these false answers (Lines 315-319) and refer the reader to a fully detailed explanation in the appendix. Kindly see our statement “we provide a full explanation of it in Appendix B.1 and Figure B.1 in the same appendix.” (Lines 317-318).
>
> **Question 6.** What is the explanation that sometimes when we increase the number of shots, the model performance drops (ex: sarcasm from 8-shot to 16-shot)? Is there anything in the settings or the prompts?
>
> **Answer 6.** Although this is probably counter-intuitive, few-shot learning can be sensitive to the order of shots as documented in the literature. This can in some cases mean that increasing the number of shots will not necessarily result in better model performance.  We emphasize that we pointed this behavior in the paper upon discussing results of the Autocompletions task. That is, in in lines 302-305 we state:  “We notice a drop in model performance under the 24-shot setting, perhaps since few-shot learning can be sensitive to the order of the shots (Wei et al., 2021; Brown et al., 2020; Lu et al., 2022).” We add that this behavior can also be related the quality of the shots, the length of the samples provided, etc. We refer to the sources cited here.
>
>
> **Question 7.** Adding SOTA results is very useful in the tables
>
> **Answer 7.** We extensively assessed JASMINE across 23 varied datasets, covering Perplexity (6 datasets), Autocompletion (3 datasets), Commonsense Inference (1 dataset), Word Manipulation (7 datasets), and natural language understanding (6 datasets). Notably, 16 out of these 23 datasets are ones that we newly create in this work. In addition, for the six perplexity datasets, we pick a total of 6k instances (1K from each of 6 domains) from publicly available sources not previously used. We needed to create these new datasets in order to provide comprehensive evaluations of our models in this area our work is pioneering. Since these 16 datasets are new, this also means there are no SOTA systems on them that we could compare to.
> Notably, SOTA comparisons are viable only seven datasets. These are the word manipulation task from QALB (Zaghouani et al., 2014) and the six NLU tasks from ORCA (Elmadany et al., 2023). We emphasize that we will incorporate SOTA results from these works in our camera-ready.
>
>
>
> **Question 8.** Prompts in tweets are very short, ex: "tweet: who/we/how...". Judging the quality of the generated outputs in such cases has no meaning.
>
> **Answer 8.** Since tweets themselves are very short in nature, we kept the prompts we used brief. In spite of this shortness, we emphasize that our model has been able to generate sensible sequences that look like tweets as illustrated in Table 7 (between Lines 383-384). We show through human evaluation that outputs from the model under these prompts are indistinguishable by humans from real tweets. More precisely, humans are able to detect `tweets` generated by the model in only 48.53% of the time (Lines 424-426).
> For more intuition, we perceive the function of the prompt here as similar to the prefix used in text-to-text Transformer models (T5), where the goal is just to alert the model that it should output a sequence of particular nature (a tweet in our case). In summary, we believe that since the use of the prompt was successful in achieving our target goal, shortness of the prompt should not be an issue.
>
>
> **Question 9.** The criterion of the selection of the occupations in Table E.1 is not clear.
>
> **Answer 9.** Our occupation list selection is driven by various factors, such as education, skills, and salary considerations. We also consider occupations that are usual in the Arab-speaking world. For example, the list comprises professions such as bus driver, lawyer, doctor, nurse, barber, teacher, and more, totaling around 100 occupations. This compilation aims to encompass a wide range of categories. We will include these details about our list of occupations in the camera-ready version.
>
> **Question 10.** "Gender, Color, Region" section is not clear, and many details are missing. "Combined with our list of 100 professions, this gives us 1,600 generated sentences of which we keep only 1,000 sentences that contain professions. Finally, we manually classify the generated sequences into one of three categories from the manually prepared set {high-wage, medium-wage, low-wage}"
>
> **Answer 10.** Regarding bias involving occupations, we note that it is not common in most of the Arab world to find publicly available census data about compensation. In view of the lack of this information, we rely on our best estimation in breaking the occupations down into {high-wage, medium-wage, low-wage}. This is done by two native Arabic speakers, each with a Ph.D. who is also well-travelled with extensive knowledge of the Arab world. The two individuals have 100% agreement on the categorization of the model outputs involving professions after discussion. We will  release outputs from the models that we manually label for bias with tags from the set {high-wage, medium-wage, low-wage}. We will also add details about how we break down professions by compensation in the camera-ready.
> We note that the model sometimes outputs sequences that do not contain professions. For this reason, we generate 1,600 sequences and keep only 1,000 that involve professions. These are the 1,000 samples that annotators manually label.
> Please refer to Table E.3 for examples that illustrate this analysis of bias as to gender, color, and region.
>
> **Question 11.** The model should be tested on more tasks from ORCA benchmark.
>
> **Answer 11.** While our primary focus is on NLG tasks rather than NLU tasks, we conducted experiments on six text classification datasets from the ORCA benchmark, constituting 10% of ORCA. However, we will extend our evaluation to the whole of the 60 datasets of the ORCA benchmark. We will add these evaluations to the paper. We thank the reviewer for the suggestion.
>
> **Question 12.** Where is the novelty in this paper?
>
> **Answer 12.** Although we do not propose new methods, our models and the majority of our datasets (16 out of 23) are totally novel. As we state in the paper (Lines 105-126), our contributions include: (1) introducing four GPT models for Arabic, pretrained on large and diverse datasets (412GB); (2) evaluating JASMINE extensively on 23 different datasets (16 of which are novel); including (3) evaluation of the first Arabic GPT models for bias and toxicity; and (4) we plan to release our models responsibly. In addition, our JASMINE models demonstrate novel and emergent abilities such as writing social posts, poetry, and even sensible conversations (Section_5 and Table_7). We hope that the reviewer will see these as sufficient contributions, involving novel models and evaluation datasets, for the extensive amount of work we are offering in this work.

---

### Meta-Review · Area_Chair_vzeW · 2023-09-19

**Recommendation:** 4

**Metareview:**

This paper presents JASMINE, a collection of Arabic GPT models trained on a diverse dataset encompassing both standard and dialectic Arabic. The paper includes a comprehensive description of the training procedures, datasets, and evaluation on various tasks, including human evaluations. The JASMINE models generally outperform existing models like AraGPT2 and mGPT, with only a few exceptions. The authors also discuss the inherent biases and shortcomings of the models.

Main Contributions:

The authors introduce the JASMINE models, a suite of decoder transformer models for Arabic, trained on a diverse range of data from both standard and dialectic Arabic.
The paper provides a comprehensive description of the training procedures, datasets used, and evaluation on various tasks, including human evaluations.
The JASMINE models generally outperform existing models like AraGPT2 and mGPT, advancing research on Arabic generative models.
The authors also discuss the inherent biases and shortcomings of the models, providing a candid assessment of their performance.

Reasons for Acceptance:

The paper presents a major contribution to the field of Arabic generative models, advancing recent advancements of language modeling and in-context learning beyond the English language.
The JASMINE models generally outperform existing models and are tested on a diverse test set, demonstrating their robustness.
The comprehensive details and examples provided in the paper enhance its clarity and applicability.
The availability of these models and benchmarks will significantly advance research on Arabic generative models.

---

### Decision · Program_Chairs · 2023-10-07

**Decision:**

Accept-Main

**Comment:**

This paper presents JASMINE, a collection of Arabic GPT models trained on a diverse dataset encompassing both standard and dialectic Arabic. The paper includes a comprehensive description of the training procedures, datasets, and evaluation on various tasks, including human evaluations. The JASMINE models generally outperform existing models like AraGPT2 and mGPT, with only a few exceptions. The authors also discuss the inherent biases and shortcomings of the models.

Main Contributions:

The authors introduce the JASMINE models, a suite of decoder transformer models for Arabic, trained on a diverse range of data from both standard and dialectic Arabic.
The paper provides a comprehensive description of the training procedures, datasets used, and evaluation on various tasks, including human evaluations.
The JASMINE models generally outperform existing models like AraGPT2 and mGPT, advancing research on Arabic generative models.
The authors also discuss the inherent biases and shortcomings of the models, providing a candid assessment of their performance.

Reasons for Acceptance:

The paper presents a major contribution to the field of Arabic generative models, advancing recent advancements of language modeling and in-context learning beyond the English language.
The JASMINE models generally outperform existing models and are tested on a diverse test set, demonstrating their robustness.
The comprehensive details and examples provided in the paper enhance its clarity and applicability.
The availability of these models and benchmarks will significantly advance research on Arabic generative models.